# Targeted antiviral treatment using non-ionizing radiation therapy for SARS-CoV-2 and viral pandemics preparedness: Technique, methods and practical notes for clinical application

**Ayan Barbora** *, **Refael Minnes** *

Department of Physics, Faculty of Natural Sciences, Ariel University, Ariel, Israel

* ayanb@ariel.ac.il (AB); refaelm@ariel.ac.il (RM)

## Abstract

**Data Availability Statement:** All relevant data are within the manuscript and its Supporting information files.

### Objective

Pandemic outbreaks necessitate effective responses to rapidly mitigate and control the spread of disease and eliminate the causative organism(s). While conventional chemical and biological solutions to these challenges are characteristically slow to develop and reach public availability; recent advances in device components operating at Super High Frequency (SHF) bands (3–30 GHz) of the electromagnetic spectrum enable novel approaches to such problems.

### Methods

Based on experimentally documented evidence, a clinically relevant *in situ* radiation procedure to reduce viral loads in patients is devised and presented. Adapted to the currently available medical device technology to cause viral membrane fracture, this procedure selectively inactivates virus particles by forced oscillations arising from Structure Resonant Energy Transfer (SRET) thereby reducing infectivity and disease progression.

### Results

Effective resonant frequencies for pleiomorphic Coronavirus SARS-CoV-2 is calculated to be in the 10–17 GHz range. Using the relation $y = -3.308x + 42.9$ with $x$ and $y$ representing $log_{10}$ number of virus particles and the clinical throat swab Ct value respectively; *in situ* patient–specific exposure duration of ~$15x$ minutes can be utilized to inactivate up to 100% of virus particles in the throat-lung lining, using an irradiation dose of 14.5 ± 1 W/m$^2$; which is within the 200 W/m$^2$ safety standard stipulated by the International Commission on Non-Ionizing Radiation Protection (ICNIRP).

**Funding:** The author(s) received no specific funding for this work.

**Competing interests:** The authors have declared that no competing interests exist.

## Conclusions

The treatment is designed to make patients less contagious enhancing faster recoveries and enabling timely control of a spreading pandemic.

## Advances in knowledge

The article provides practically applicable parameters for effective clinical adaptation of this technique to the current pandemic at different levels of healthcare infrastructure and disease prevention besides enabling rapid future viral pandemics response.

## Introduction

Viruses account for more catastrophic epidemics than other pathogens. Emerging new infective species necessitate development of rapid responses for mitigation and disease control. Conventional responses involving medication(s) providing a treatment through selective biochemical targeting also drive disease resistance [1] and immunocompromise in the long run; further aggravating overall outcomes of healthcare in disease outbreaks. Effective long-term vaccine/drug development targeting viral components becomes a costly, difficult, sometimes impossible challenge as viruses mutate fast changing their characteristic proteins. This translates into substantially long developmental time periods before reaching public availability while rising mortality rates convert an initial epidemic into a pandemic [2].

Virus particles production starts subsequent to a cell getting infected and released virions proceed to infect other cells. The rate of cellular infections over time versus the number of uninfected cells remaining [3] and time for developing the natural immune response determine the establishment of infection/disease and its corresponding recovery. Similarly, the spread of a pandemic is analogous to game theoretical simulations of mortality rates and infected/contagious population versus recovery rates and uninfected/immunized population. SARS-CoV-2 virus presents a 10 hours eclipse period between the time of cellular viral entry and release of new particles with a typical burst size of $\sim 10^3$ virions [4]. Clinical cases indicate viral loads reduce significantly over the first few weeks of symptoms [5] with the natural antibody response developing 10–20 days post infection [4]. Thus, an effective radiation therapy for inactivating sufficient number of viral particles within the eclipse period will help to keep viral loads low giving the body's natural immune response the timely competitive edge to fight the infection.

For enveloped viruses (viruses that are surrounded by a continuous bilayer membrane studded with viral proteins), such as the SARS-CoV-2 virus, the biophysical properties of the envelope are determined according to the composition of the membrane and its dimensions [6–8]. For such viruses, the infectivity of the virus is physically inactivated by membrane lysis. Viral genomes released from solubilized/ruptured membranes in itself remains incapable of infecting eukaryotic cells as demonstrated by the historically successful disease control afforded by alcohol, soap, etc. [9, 10]. Overcoming evolution of disease/drug resistance, such methods hold proven effectiveness till date [11]. However, alcohol or soap cannot be directly applied within the body to inactivate virions during shedding of new particles from infected tissue linings. Instead, a membrane rupture radiation technique for selective viral particles inactivation will be highly effective for rapid treatment enhancing faster recoveries and controlling the spread of a pandemic.

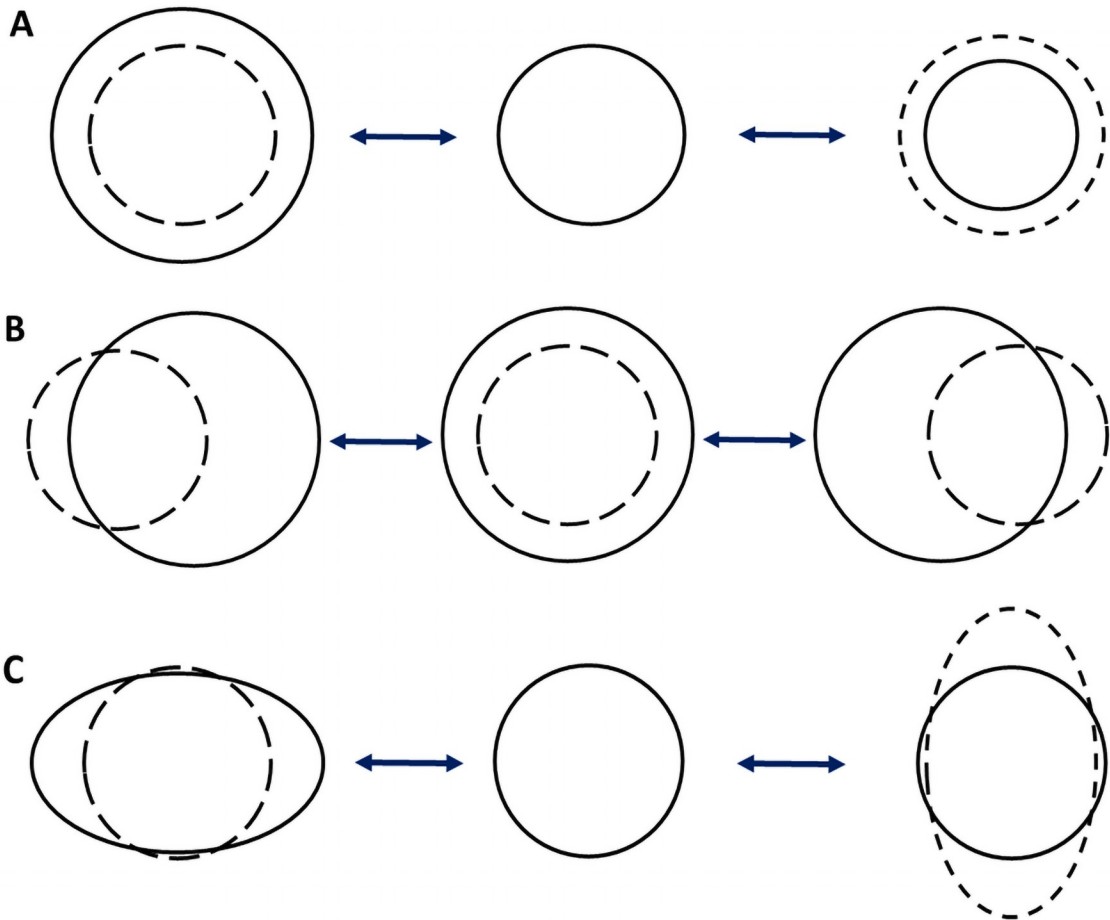

**Fig 1. Illustration of oscillations in spherical particles with core-shell charge separations.** (A) monopolar, (B) dipolar and (C) quadrupolar modes. The dotted and undotted lines represent relative opposite charge densities respectively.

Low-Dose Radiation Therapy (LDRT) has been tried for treating SARS-CoV-2 [12]. Ionizing radiation degrading all exposed biochemical substrate(s) [13], the therapeutic effects of this method may arise from viral inactivation besides exposure associated cytotoxic effects killing off virus infected cells, limiting the widespread application of LDRT. Instead a radiation technique which allows selective viral inactivation only without cytotoxicity will be a safer and far more effective therapy to implement universally. Nano dimensional condensed matter with core-shell charge separations like spherical viruses present dipolar mode oscillatory vibrations (Fig 1B) upon electromagnetic coupling with resonant frequencies [14] within the non-ionizing radiation bands. Particle dimensions and pH determine the resonant frequency and power absorption respectively. This Structure Resonant Energy Transfer (SRET) phenomenon elicits Confined Acoustic Vibrations (CAV) wherein opposite core-shell oscillations result in physical viral membrane fracture providing effective inactivation [15].

Non-ionizing radiation (1–300 GHz) with lesser photon energy incapable of causing ionization or breaking chemical bonds unlike ionizing radiation, generates only thermal stresses at certain power densities [16]. And such thermal stresses are alleviated using appropriate tuning and exposure conditions to elicit non-thermal biological responses [15–17] for medical

applications. Although non-ionizing radiation penetration into the body is low compared to ionizing radiation exposure from external sources, recent advances in endoscopic Super High Frequency (SHF) (3–30 GHz) medical devices [18] allow delivery of such radiation to deep tissues within the human body. These developments encourage the adaptation of this antiviral method [15] for viral pandemic(s) response. SARS-CoV-2 viruses start replicating and shedding in the throat even before reaching the lungs [19]. Therefore, an antiviral throat/lung treatment procedure of non-thermal SRET mediated inactivation [15] using biomedical non-ionizing radiation devices [18] and techniques [14–17] for reducing viral loads will be highly effective for critically ill patients.

Coronavirus SARS-CoV-2 resonant frequencies are estimated in this article and a method presented to determine patient-specific therapeutic exposure duration(s) versus corresponding number of inactivated particles based on standard clinical throat swabs; within the ICNIRP (International Commission on Non-Ionizing Radiation Protection) dosage safety standards. A practical method to apply this technique for clinical adaptation at different levels of pandemic response and disease prevention for other viruses in the future is also described.

## Methods

### Dipolar mode spherical oscillations

Using Lamb's theory [20], frequency of the dipolar mode ($l = 1$, $n = 0$) for any spherical particle (Fig 1) can be estimated using the eigen value equation:

$$4\frac{j_2(\xi)}{j_1(\xi)} - \eta^2 + 2\frac{j_2(\eta)}{j_1(\eta)}\eta = 0$$

where $\xi = \frac{2\pi vD}{V_L}$, $\eta = \frac{2\pi vD}{V_T}$ and $j_l(\eta)$ is the spherical Bessel function of the first kind. D is the diameter, $V_L$ and $V_T$ denote longitudinal and transverse sound velocities of the particle respectively.

Solving this equation (ref. S1 File) for resonant frequency simplifies it to

$$f = \frac{V_L}{2D} \tag{1}$$

### Virus charge status

Absorption cross-section of a virus $\sigma_{abs} = \frac{Qq^2}{\omega_0 m^* c \sqrt{\varepsilon_r}\varepsilon_0}$ [15]; where $Q$, $q$, $\omega_0$, $m^*$, $c$, $\varepsilon_r$ and $\varepsilon_0$ denote quality factor of the resonator, total amount of charge, intrinsic resonant angular frequency, reduced mass of core—shell virus particle, speed of light in vacuum, relative permittivity in the system and the absolute permittivity of free space respectively. Constant $c$ remaining unchanged the charge status of two viruses can be related by

$$\frac{Q_1 q_1^2}{\omega_{0_1} m_1^* \sigma_{abs_1}\sqrt{\varepsilon_{r_1}}\varepsilon_0} = \frac{Q_2 q_2^2}{\omega_{0_2} m_2^* \sigma_{abs_2}\sqrt{\varepsilon_{r_2}}\varepsilon_0} \tag{2}$$

with the numbers 1 and 2 representing the respective viruses.

### Threshold electric field intensities to fracture viral particles

Threshold electric field magnitude $E_T$ to cause viral inactivation is estimated using

$$E_T = \frac{P_{Stress}^T \ \pi r^2 \sqrt{m^* \left( \omega_0^2 - \omega^2 \right) + \left( \frac{\omega_0 m^*}{Q} \right)^2 (\omega^2)}}{3.45 \ q m^* \omega_0^2}$$

[15]

where $P_{Stress}^T$ and $r$ denote the threshold stress to fracture the membrane and the radius respectively while $m^*$, $\omega_0$, $q$ and $Q$ represent the same parameters described above. With $\frac{\pi}{3.45}$ remaining a constant; the respective parameters for two distinct viruses can be related by

$$\frac{P_{Stress_1}^T \ r^2_{1} \sqrt{m^*_{1} \left( \omega_{0_1}^2 - \omega^2_{1} \right) + \left( \frac{\omega_{0_1} m^*_{1}}{Q_1} \right)^2 (\omega^2_{1})}}{E_{T_1} q_1 m^*_{1} \omega_{0_1}^2}$$
$$= \frac{P_{Stress_2}^T \ r^2_{2} \sqrt{m^*_{2} \left( \omega_{0_2}^2 - \omega^2_{2} \right) + \left( \frac{\omega_{0_2} m^*_{2}}{Q_2} \right)^2 (\omega^2_{2})}}{E_{T_2} q_2 m^*_{2} \omega_{0_2}^2} \tag{3}$$

with the numbers 1 and 2 representing the respective viruses.

## Results

### Resonant frequencies for SARS-CoV-2

To estimate resonant frequencies for SARS-CoV-2, parameters of diameter and longitudinal sound velocity of the viral membrane are required [14]. To ascertain a practically applicable particle dimension index and sound velocity for any spherical virus using Lamb's Theory [14, 15, 20]; calculated resonant frequencies for different spherical particle dimensions of EV71 and sound velocities were compared with the experimental observations. Experimentally EV71 viruses with 35 ± 2 nm hydrodynamic diameter at 6.4 pH exhibits resonant dipolar coupling at ~45 GHz [14]. But electron microscopy imaging [21] denotes EV71 diameter as 28.5 ± 1.5 nm. Fig 2 presents estimated resonant frequencies for different EV71 diameters and $V_L$ range of 1800–3000 m/s based on elastic properties of viruses [22]. Poisson's ratio of most condensed matter being $\sim 0.3$ [23], $V_L/V_T$ ratio is set as 2 in the calculations deriving Eq 1 (ref. methods).

The experimental observation of ~45 GHz for EV71 corresponds to $V_L$ of 2565 m/s with 28.5 ± 1.5 nm diameter. Diameter of influenza A virus using this value of $V_L$ (2565 m/s) and the experimental resonant frequency of 12 GHz [14], derives to be 107 ± 6 nm through Eq 1 (ref. methods); while it is typically reported to be 100 nm [24] by electron microscopy. Resonant frequency for influenza A virus using 2400 m/s as $V_L$ and 93 ± 5 nm hydrodynamic diameter at pH 7.4 [14] is calculated to be 12.9 ± 0.7 GHz; while using the same $V_L$ (2400 m/s) with electron microscopy denoted 100 nm diameter [24] the resonant frequency derives to be 12 GHz matching the experimental observations [14] (Table 1). Also, resonant frequency for EV71 virus calculated using 2400 m/s as $V_L$ and 28.5 ± 1.5 nm electron microscopy denoted diameter [22], derives to be 42 ± 2 GHz matching the experimental observations [14] (Table 1). Further, the resonant frequency of EV71 remains constant at ~45 GHz although hydrodynamic diameter and power absorption varies with changing pH from 5.4–7.4 [14]. This indicates that for practical approximations electron microscopy imaging denoted diameter ($D$) in nm can be considered with 2400 m/s as $V_L$ to estimate effective resonant

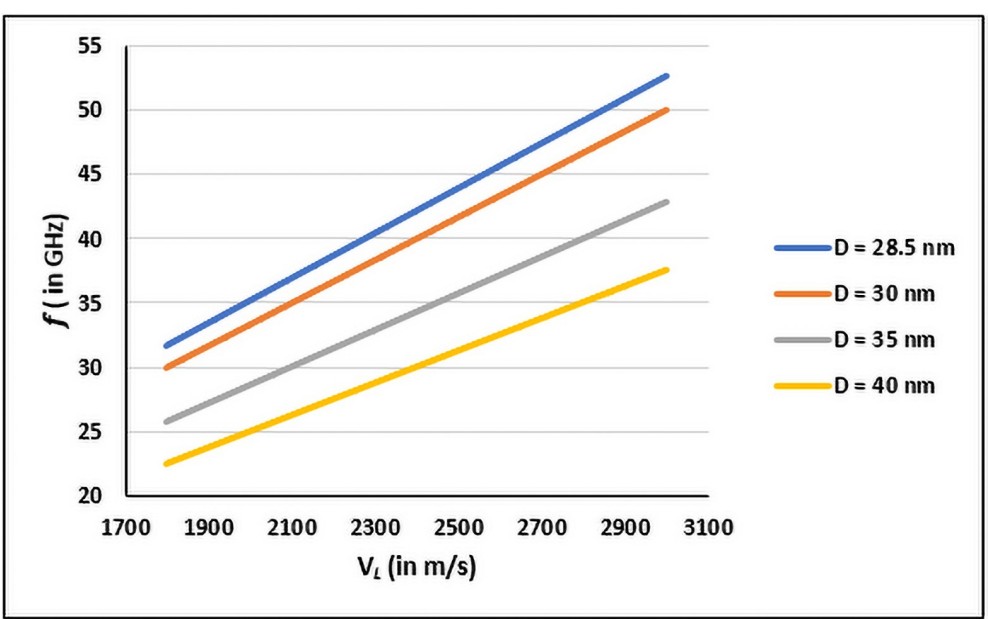

**Fig 2. Resonant frequencies $f$ (in GHz) corresponding to respective $V_L$ (in m/s) for different diameters D (in nm) of EV71 virus.**

frequencies ($f$) in GHz for spherical viral particles using

$$f = \frac{2400}{2D} \tag{4}$$

while pH determined hydrodynamic dimensions are useful for other estimates as discussed later. With diameters ranging from 60–140 nm [25] and average size of 70–80 nm [26] determined using electron microscopy, resonant frequencies for pleiomorphic SARS-CoV-2 is calculated to be within 8.5–20 GHz with absorption peak likely within 15–17 GHz as presented in Table 1.

## Power density for SARS-CoV-2 inactivation

Reduced mass $m^*$ of SARS-CoV-2 particles [4] is ~55.63 MDa (approx. 9.61 MDa from 30 kb ssRNA and 46.02 MDa from 1000 units of nucleoprotein [27]). Without Atomic Force Microscopy of SARS-CoV-2 but given its 50% genomic similarity with Common Cold CoV [4] and

**Table 1. Resonant frequencies $f$ (in GHz) corresponding to different diameters D (in nm) for influenza A, EV71 and SARS-CoV-2.**

| Virus | Diameter (in nm) | Resonant Frequency (in GHz) $f = \frac{2400}{2D}$ |
|---|---|---|
| Influenza A | 93 ± 5 (H D) [15] | 12.9 ± 0.7 |
| | 100 (EM D) [26] | 12 |
| EV71 | 35 ± 2 (H D) [15] | 34 ± 2 |
| | 28.5 ± 1.5 (EM D) [23] | 42 ± 2 |
| SARS-CoV-2 | 60–140 (EM D complete range) [25] | 8.5–20 |
| | 70–80 (EM D average size) [26] | 15–17 |

H D and EM D indicate hydrodynamic and electron microscope diameters respectively.

that viral shells derive typically from host cell membrane portions; 0.141 MPa is considered as $P_{Stress}^{T}$ in our calculations, similar to that for influenza A virus subtype H3N2 [15].

Virus dispersion conditions within the medium determines quality factor $Q$. Viruses aggregate or precipitate [28] at their characteristic isoelectric points (pI), i.e. the pH value where particles bear zero net charge. This affects power absorption for resonant electromagnetic coupling as surface charge status determines dispersion conditions, crucial for generating confined acoustic vibrations. Influenza A virus subtype H3N2 presented $Q$ of 1.95 [15] in PBS medium (pH 7.4). Considering SARS-CoV-2 being extremely stable over a wide range of pH (3–10) at room temperature [11] and pH 6.9 [29] in the lung lining fluid, a conservative estimate of $Q$ is set as 2.

Power absorption is proportional to the number of virus particles for a fixed cross-sectional area. Assuming N number of SARS-CoV-2 virus particles (ref. methods) present an absorption cross section $\sigma_{abs}$ of $2.5 \times 10^{-3}$ m$^2$ (detected similarly to the influenza A virus subtype H3N2 inactivation model [15]) and $\varepsilon_r$ for Lung lining fluid being ~20 for 15–17 GHz [30], $q$ is calculated to be $\sim 2.32 \times 10^{7} e$ using Eq 2 (ref. methods). Using these values, threshold electric field magnitude $E_T$ to cause SARS-CoV-2 virus particles inactivation exhibiting resonance at 15–17 GHz is estimated to be 24 ± 3 V/m (about 1.5 ± 0.4 W/m$^2$) using Eq 3 (ref. methods).

## Discussion

Sweeping frequency regimes are more productive than discrete frequency irradiations in eliciting corresponding biological response(s) [31, 32]. But, viral inactivation will only arise from sustained resonance at the estimated frequencies. Therefore, considering the pleiomorphic properties of SARS-CoV-2 [25], a simultaneous propagation of 10–17 GHz resonant frequencies will be the most effective.

Treatment of $7.5 \times 10^{14}$ m$^{-3}$ influenza A subtype H3N2 virus particles at the theoretical threshold of 87 V/m (82 W/m$^2$) [15] for 15 minutes corresponded to 38% inactivation while 270 V/m (820 W/m$^2$) [15] over the same exposure duration corresponded to 100% inactivation. The International Commission on Non-Ionizing Radiation Protection (ICNIRP) stipulates 200 W/m$^2$ absorbed power density for 100 kHz to 300 GHz as the non-thermal public safety standard [33]. Thus, a simultaneous 10–17 GHz irradiation can provide 38% inactivation for ~15 minutes exposure at 1.5 ± 0.4 W/m$^2$ (24 ± 3 V/m) threshold power density; and 100% inactivation for ~15 minutes exposure at 14.5 ± 1 W/m$^2$ (74 ± 3 V/m) power density. Clinical viral densities inside patients are estimated using the equation $y = -3.308x + 42.9$ [34]; wherein $x$ and $y$ denote $log_{10}$ RNA equivalents (i.e. $log_{10}$ number of particles) and throat swab Ct values [5] respectively. Therefore, patient-specific exposure durations of this non-ionizing radiation treatment can be appropriately derived as $15x$ minutes to provide 100% inactivation of up to $7.5 \times 10^{14}$ m$^{-3}$ SARS-CoV-2 virus particles at 14.5 ± 1 W/m$^2$.

Non-thermal SRET induced influenza A subtype H3N2 and H1N1 viral membrane rupture was confirmed to leave viral genomes intact unlike microwave heating [15] and the unablated membrane-free genome did not generate further infection. Cells tolerate 50–150 W/m$^2$ power densities of non-ionizing radiation during therapeutic procedures [35] without developing cellular stress or thermal effects. Resonance effects at the same power density elicit distinct biological responses from different samples accountable to their distinct biochemical composition (s) and membrane associated charge separation(s). Illustratively, 2 W/m$^2$ W—band EHF exposure for 10 minutes caused mortality of H1299 cancer cells while the same number of normal MCF-10A cells remained unaffected even after 16 minutes irradiation under similar conditions [31]. Epithelial cells tolerate irradiation at 10 W/m$^2$ over 24 hours of exposure without developing genotoxicity or thermal stress [36]. Therefore, the estimate of 14.5 ± 1 W/m$^2$

should be effective in selectively targeting SARS-CoV-2 virus particles without generating therapy associated stresses on patients receiving such treatment.

Conventional pharmaceutical/biological remedies being biochemical solutions are subject to evolution of mutant strains often rendering them ineffective, thereby necessitating constant production of stronger antidotes besides driving drug resistance. In contrast the non-ionizing radiation antiviral treatment therapy presented here works on resonant frequency and appropriate power dose [14, 15] determined solely by the physical particle dimensions of the viruses. As such, this technique is advantageous over other treatments given its dynamic adaptability which allows tuning the radiation parameters to particle dimensions of any virus concerned. Further, being a radiation-based therapy, this treatment can effectively reach deep tissues endoscopically [18], wherein any and all virus particles within the targeted resonant frequency range [14, 15] get their membranes ruptured during irradiation as soon as they form or are lying dormant. And viruses with membranes ruptured via this SRET based method are demonstrated to be incapable of causing further infection upon introduction to other non-infected cells [15].

## Conclusions

The equation $y = -3.308x + 42.9$ [34]; provides a relation between $log_{10}$ number of virus particles and clinical throat swab Ct values represented by $x$ and $y$ respectively. Patient-specific exposure durations of ~15$x$ minutes can be derived for providing 100% inactivation of SARS-CoV-2 virus particles by 10–17 GHz simultaneous irradiations at 14.5 ± 1 W/m$^2$. For the SARS-CoV-2 pandemic with an eclipse period of 10 hours between cellular entry and the release of new particles with a typical burst size of $\sim 10^3$ virions [4] this method can reduce viral loads significantly in symptomatic as well as asymptomatic patients enhancing faster recoveries and reducing pandemic mortality rates. Coupled with recent advances in rapid diagnostic procedures [37, 38] this *in situ* antiviral treatment can rapidly improve patient outcome(s) enabling control over the spreading pandemic without necessitating economically adverse regional lockdowns [39]. With the natural antibody response against SARS-CoV-2 developing 10–20 days post infection [4], this therapy can provide the timely competitive edge for the body's immunity to fight infection and aid patient recoveries by keeping the viral loads low.

The treatment method described here can also be used for air purification, sanitizing public spaces, health care centers, travel/transport gateways, etc. by surface swabbing for viral particle densities and appropriately scaling the exposure conditions corresponding to such environments using Eqs 1–4 presented above.

The greatest advantage of this technique is its ready applicability across the globe as a treatment against the same disease once characteristic parameters of a virus for this technique are determined. Also, by allowing adequate modification of the exposure conditions using Eqs 1–4 for the targeted viruses respectively, this technique provides an inherent tunability for treating any new and emerging viral epidemic. Further, this study encourages rapid physical characterization techniques of novel viruses as and when they appear through Electron Microscopy, Atomic Force Microscopy, Chemical Force Microscopy [40] etc. methods in order to determine corresponding dosimetry of clinical non-ionizing radiation therapy; providing rapid solutions for targeted antiviral treatments using advanced clinical SHF medical devices [18].

## Supporting information

**S1 File.**
(DOCX)

## Acknowledgments

Ayan Barbora gratefully acknowledges useful discussions with Prof. Asher Yahalom of the Israel Free Electron Laser Centre, Dr. D. Sarkar and Mr. S. Shrivastava of the Tata Institute of Fundamental Research.

## Author Contributions

**Conceptualization:** Ayan Barbora, Refael Minnes.

**Data curation:** Ayan Barbora.

**Formal analysis:** Ayan Barbora, Refael Minnes.

**Investigation:** Ayan Barbora.

**Methodology:** Ayan Barbora, Refael Minnes.

**Software:** Ayan Barbora.

**Validation:** Refael Minnes.

**Visualization:** Ayan Barbora.

**Writing – original draft:** Ayan Barbora, Refael Minnes.

**Writing – review & editing:** Ayan Barbora, Refael Minnes.

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
