## [Decision Letter · Decision Letter 0]

25 Mar 2021

PONE-D-20-31505

Targeted antiviral treatment using Non-ionizing Radiation Therapy for SARS-CoV-2 and viral pandemics preparedness: Technique, methods and practical notes for Clinical Application

PLOS ONE

Dear Dr. Minnes,

Thank you for submitting your manuscript to PLOS ONE. After careful consideration, we feel that it has merit but does not fully meet PLOS ONE’s publication criteria as it currently stands. Therefore, we invite you to submit a revised version of the manuscript that addresses the points raised during the review process.

We look forward to receiving your revised manuscript.

Kind regards,

Balveen Kaur

Academic Editor

PLOS ONE

Journal Requirements:

Reviewers' comments:

Reviewer's Responses to Questions

**Comments to the Author**

1. Is the manuscript technically sound, and do the data support the conclusions?

Reviewer #1: Yes

Reviewer #2: Partly

2. Has the statistical analysis been performed appropriately and rigorously? 

Reviewer #1: No

Reviewer #2: I Don't Know

3. Have the authors made all data underlying the findings in their manuscript fully available?

Reviewer #1: Yes

Reviewer #2: Yes

4. Is the manuscript presented in an intelligible fashion and written in standard English?

Reviewer #1: Yes

Reviewer #2: Yes

5. Review Comments to the Author

Reviewer #1: The method part should be detailed. theoretical information should be made understandable by giving examples. The type of radiation to be applied should be given clearly. Other parts of the article contain very nice and useful information.

Reviewer #2: Ayan Barbora and colleague present the feasible application of Non-ionizing Radiation therapy for SARS-COV-2 treatment. This study addresses an interesting aspect of SARS-COV-2 treatment during the pandemic outbreak. The authors present data of Non-Ionizing Radiation therapy in the treatment of EV71 and influenza A to devise the theoretical condition for SARS-COV-2 treatment using SRET.

Major point:

- The novel treatment for SARS-COV-2 by using Non-ionizing Radiation Therapy is calculated from the procedure used in the treatment of EV71 and influenza virus. If the author can test the procedure in SARS-COV-2, the conclusion will be more convincing.

Minor points:

- The authors devise a new treatment based on the virus physical properties, such as the virus’ particle size and charge. Does SARS-COV-2 biological characteristics, such as anatomical site of virus entry and replication, affect the efficacy of the new procedure proposed in this study? The authors should discuss this aspect in the discussion section.

6. PLOS authors have the option to publish the peer review history of their article (what does this mean?). If published, this will include your full peer review and any attached files.

Reviewer #1: No

Reviewer #2: No

---

## [Author Response · Author response to Decision Letter 0]

8 Apr 2021

Point-by-Point Response to Reviewers and Editor

In the following, we address all the reviewers’ and editor’s comments:

Reviewers’ and editor’s comments in red, our response in black.

Reviewer #1: 

The method part should be detailed. theoretical information should be made understandable by giving examples. The type of radiation to be applied should be given clearly. Other parts of the article contain very nice and useful information.

Our article presents a novel and fast antiviral treatment method that was developed based on published experimental evidence. Since the PLOS One guidelines demand that the Methods section come before the Results section, kindly refer to:

1. Results section containing the documented experimental examples, background information, theoretical basis of our technique and the type (frequency) and power dosage of radiation to be used; and

2. Methods section containing the mathematical formulations and derivatives

3. Discussion section containing details on how clinicians can derive appropriate treatment durations of the radiation involved by using the standard Ct values from viral swab tests done universally today.

To make our argument clearer, we added some points in Introduction. 

We rewrote the following: 

"For enveloped viruses (viruses that are surrounded by a continuous bilayer membrane studded with viral proteins), such as the SARS-CoV-2 virus, the biophysical properties of the envelope are determined according to the composition of the membrane and its dimensions [6, 7, 8]. For such viruses, the infectivity of the virus is physically inactivated by membrane lysis. Viral genomes released from solubilized/ruptured membranes in itself remains incapable of infecting eukaryotic cells as demonstrated by the historically successful disease control afforded by alcohol, soap, etc. [9, 10]."

and: 

"Although non-ionizing radiation penetration into the body is low compared to ionizing radiation exposure from external sources, recent advances in endoscopic Super High Frequency (SHF) (3-30 GHz) medical devices [18] allow delivery of such radiation deep within the human body. Such developments encourage the adaptation of this antiviral method [15] for viral pandemic(s) response."

Details on the type and doses of the radiation can be found in Conclusions:

" The equation y = -3.308x + 42.9 [3134]; provides a relation between 〖log〗_10 number of virus particles and clinical throat swab Ct values represented by x and y respectively. Patient-specific exposure durations of ~15x minutes can be derived for providing 100 % inactivation of SARS-CoV-2 virus particles by 10 – 17 GHz simultaneous irradiations at 14.5 ± 1 W/m2. For SARS-CoV-2 pandemic with an eclipse period of 10 hours between cellular entry and the release of new particles with a typical burst size of ∼103 virions [4] this method can reduce viral loads significantly enhancing patient recoveries and reducing pandemic mortality rates."

Reviewer #2: 

Ayan Barbora and colleague present the feasible application of Non-ionizing Radiation therapy for SARS-COV-2 treatment. This study addresses an interesting aspect of SARS-COV-2 treatment during the pandemic outbreak. The authors present data of Non-Ionizing Radiation therapy in the treatment of EV71 and influenza A to devise the theoretical condition for SARS-COV-2 treatment using SRET.

Major point:

- The novel treatment for SARS-COV-2 by using Non-ionizing Radiation Therapy is calculated from the procedure used in the treatment of EV71 and influenza virus. If the author can test the procedure in SARS-COV-2, the conclusion will be more convincing.

We present an innovative method of in situ antiviral treatment based on the physical phenomenon of SRET which has been experimentally demonstrated to be highly effective in inactivating other viruses previously (ref. EV71 and influenza virus examples). Being a physical phenomenon based on particle dimensions it is directly applicable with the currently available medical device technologies (ref. 18). However, since we currently do not have authorization/access to work with SARS-COV-2 viruses we are unable to test it out. 

Minor points:

- The authors devise a new treatment based on the virus physical properties, such as the virus’ particle size and charge. Does SARS-COV-2 biological characteristics, such as anatomical site of virus entry and replication, affect the efficacy of the new procedure proposed in this study? The authors should discuss this aspect in the discussion section.

We have added a new paragraph addressing these issues in the Discussion: 

" Conventional pharmaceutical/biological remedies being biochemical solutions are subject to evolution of mutant strains often rendering them ineffective, thereby necessitating constant production of stronger antidotes besides driving drug resistance. In contrast the non-ionizing radiation therapy antiviral treatment presented here works on resonant frequency and appropriate power dose [14, 15] determined solely by the physical particle dimensions of the viruses. As such, this technique is advantageous over other treatments given its dynamic adaptability which allows tuning the radiation parameters to particle dimensions of any virus concerned. Further, being a radiation-based therapy, this treatment can effectively reach deep tissues endoscopically [18], wherein any and all virus particles within the targeted resonant frequency range [14, 15] get their membranes ruptured during irradiation as soon as they form or are lying dormant. And viruses with membranes ruptured via this SRET based method are incapable of causing further infection upon introduction to other non-infected cells [15]."

PLOS One Comments:

https://journals.plos.org/plosone/s/file?id=ba62/PLOSOne_formatting_sample_titleauthors_affiliations.pdf

The manuscript's style was changed according to the style requirements of PLOS ONE. 

We would like to thank the reviewers for their comments. We honestly believe that these comments helped us improve our manuscript. 

Sincerely,

Dr. Refael Minnes (corresponding author)

Department of Physics, Ariel University. 

40700, Ariel, Israel

Cell: (972)-54-6254094

Office: (972)-3-6453140

Email: refaelm@ariel.ac.il

---

## [Decision Letter · Decision Letter 1]

4 May 2021

Targeted antiviral treatment using Non-ionizing Radiation Therapy for SARS-CoV-2 and viral pandemics preparedness: Technique, methods and practical notes for Clinical Application

PONE-D-20-31505R1

Dear Dr. Minnes,

We’re pleased to inform you that your manuscript has been judged scientifically suitable for publication and will be formally accepted for publication once it meets all outstanding technical requirements.

Kind regards,

Eric Charles Dykeman, Ph.D.

Academic Editor

PLOS ONE

Additional Editor Comments (optional):

Reviewers' comments:

Reviewer's Responses to Questions

**Comments to the Author**

1. If the authors have adequately addressed your comments raised in a previous round of review and you feel that this manuscript is now acceptable for publication, you may indicate that here to bypass the “Comments to the Author” section, enter your conflict of interest statement in the “Confidential to Editor” section, and submit your "Accept" recommendation.

Reviewer #1: All comments have been addressed

Reviewer #2: All comments have been addressed

2. Is the manuscript technically sound, and do the data support the conclusions?

Reviewer #1: Yes

Reviewer #2: (No Response)

3. Has the statistical analysis been performed appropriately and rigorously? 

Reviewer #1: Yes

Reviewer #2: (No Response)

4. Have the authors made all data underlying the findings in their manuscript fully available?

Reviewer #1: Yes

Reviewer #2: (No Response)

5. Is the manuscript presented in an intelligible fashion and written in standard English?

Reviewer #1: Yes

Reviewer #2: (No Response)

6. Review Comments to the Author

Reviewer #1: All corrections were made. This article can be accepted.

Although non-ionizing radiation penetration into the body is low compared to ionizing

radiation exposure from external sources, recent advances in endoscopic Super High

Frequency (SHF) (3-30 GHz) medical devices [18] allow delivery of such radiation

deep within the human body. Such developments encourage the adaptation of this

antiviral method [15] for viral pandemic(s) response."

Details on the type and doses of the radiation can be found in Conclusions:

" The equation y = -3.308x + 42.9 [3134]; provides a relation between 〖log〗_10

number of virus particles and clinical throat swab Ct values represented by x and y

respectively. Patient-specific exposure durations of ~15x minutes can be derived for

providing 100 % inactivation of SARS-CoV-2 virus particles by 10 – 17 GHz

simultaneous irradiations at 14.5 ± 1 W/m2. For SARS-CoV-2 pandemic with an

eclipse period of 10 hours between cellular entry and the release of new particles with

a typical burst size of ∼103 virions [4] this method can reduce viral loads significantly

enhancing patient recoveries and reducing pandemic mortality rates."

2.Methods section containing the mathematical formulations and derivatives

3.Discussion section containing details on how clinicians can derive appropriate

treatment durations of the radiation involved by using the standard Ct values from viral

swab tests done universally today.

To make our argument clearer, we added some points in Introduction.

Reviewer #2: (No Response)

7. PLOS authors have the option to publish the peer review history of their article (what does this mean?). If published, this will include your full peer review and any attached files.

Reviewer #1: **Yes: **Suleyman GOKMEN

Reviewer #2: **Yes: **Bangxing Hong

---

## [Editor Report · Acceptance letter]

6 May 2021

PONE-D-20-31505R1 

Targeted antiviral treatment using Non-ionizing Radiation Therapy for SARS-CoV-2 and viral pandemics preparedness: Technique, methods and practical notes for Clinical Application 

Dear Dr. Minnes:

I'm pleased to inform you that your manuscript has been deemed suitable for publication in PLOS ONE. Congratulations! Your manuscript is now with our production department. 

Kind regards, 

on behalf of

Dr. Eric Charles Dykeman 

Academic Editor

PLOS ONE